

# Projections of coral reef carbonate production from a global climate-coral reef coupled model

Nathaelle Bouttes[1], Lester Kwiatkowski[2], Elodie Bougeot[1], Manon Berger[3], Victor Brovkin[4], Guy Munhoven[5]

[1]Laboratoire des Sciences du Climat et de l'Environnement, LSCE/IPSL, CEA ‑ CNRS ‑ UVSQ, Université Paris ‑ Saclay, Gif ‑ sur ‑ Yvette, 91191, France

[2]LOCEAN Laboratory, Sorbonne Université-CNRS-IRD-MNHN, Paris, 75005, France
[3]LMD-IPSL, CNRS, Ecole Normale Supérieure/PSL Res. Univ, Ecole Polytechnique, Sorbonne Université, Paris, 75005, France
[4]Max Planck Institute for Meteorology, Hamburg, 20146, Germany; also CEN, University of Hamburg, Hamburg, 20146, Germany;
[5]Dépt. d'Astrophysique, de Géophysique et d'Océanographie, Université de Liège, Liège, B-4000, Belgium

*Correspondence to*: Nathaelle Bouttes (Nathaelle.bouttes@lsce.ipsl.fr)

**Abstract.** Coral reefs are under threat due to climate change and ocean acidification. However, large uncertainties remain concerning future carbon dioxide emissions, climate change and the associated impacts on coral reefs. While most previous studies have used climate model outputs to compute future coral reef carbonate production, we use a coral reef carbonate production module embedded in a global carbon-climate model. This enables the simulation of the response of coral reefs to projected changes in physical and chemical conditions at finer temporal resolution. The use of a fast-intermediate complexity model also permits the simulation of a large range of possible futures by considering different greenhouse gas concentration scenarios (Shared Socioeconomic Pathways (SSPs)), different climate sensitivities (hence different levels of warming for a given level of acidification), as well as the possibility of corals adapting their thermal bleaching thresholds. We show that without thermal adaptation, global coral reef carbonate production decreases to less than 25% of historical values in most scenarios over the twenty-first century, with limited further declines between 2100 and 2300 irrespective of the climate sensitivity. With thermal adaptation, there is far greater scenario variability in projections of reef carbonate production. Under high-emission scenarios the rate of twenty-first century declines is attenuated, with some global carbonate production declines delayed until the twenty-second century. Under high-mitigation scenarios, however, global coral reef carbonate production can recover in the twenty-first and twenty-second century, and thereafter persists at 50-90% of historical values, provided that the climate sensitivity is moderate.



## 1 Introduction

Coral reefs are marine ecosystems composed of reef-building corals. The corals build structures (exoskeletons) made of calcium carbonate in the form of aragonite. The carbonate structures not only provide a habitat for many marine species, but also play a role in the carbon cycle. Indeed, the production of calcium carbonate modifies the carbonate equilibrium in the ocean, which can ultimately modify the atmospheric $CO_2$ concentration and impact climate. Understanding and modelling carbonate production is therefore crucial for both quantifying reefs impacts and anticipating possible climate feedbacks.


While coral reefs are among the most diverse and valuable ecosystems on Earth, they are under multiple threats due to human induced changes on their environment (Pandolfi et al., 2011; Hoegh-Guldberg et al., 2017). With increasing atmospheric carbon dioxide concentrations, marine organisms face ocean warming and ocean acidification. Climate models indicate that under the high-emission scenario SSP5-8.5, sea surface temperatures will increase by $3.47 \pm 0.78$ °C, while the surface pH

will decrease by $-0.44 \pm 0.005$ by the end of the century (multi-model global mean change values of 2080–2099 relative to 1870–1899 $\pm$ one standard deviation, Kwiatkowski et al, 2020). With the low-emission scenario SSP1-2.6, SSTs are still projected to increase, albeit to a lesser extent, by $1.42 \pm 0.32$ °C. The surface pH decrease is also smaller, of $-0.16 \pm 0.002$. Not only will temperatures increase, but marine heatwaves will become longer-lasting and more frequent (Frölicher et al., 2018).


Both ocean warming and acidification have negative effects on coral reefs. Increased seawater temperatures, and in particular increased marine heat wave frequency and intensity, is damaging for coral reefs (Cooley et al., 2022). Under heat stress, coral polyps expel their zooxanthellae symbionts, resulting in coral bleaching (Brown, 1997; Hoegh-Guldberg, 1999; Sully et al., 2019). Depending on the rate at which seawater temperatures return to climatological levels of the early and mid 20[th] century,

corals can recover and zooxanthellae symbiosis can resume (DeCarlo et al., 2019; Logan et al., 2021). However, under extreme or repeated heat stress, bleaching becomes severe and can lead to the coral death.

Due to reduced pH and carbonate ion concentration, the calcification process which enables organism to produce their external calcium carbonate skeleton requires more energy. To characterise the carbonate ion concentration, we consider the aragonite

saturation state ($\Omega_{ar}$) defined as the ratio of the product of calcium and carbonate ion concentrations to the equilibrium thermodynamic solubility product ($K_{sp}$) for the mineral aragonite at in-situ temperature, salinity and pressure:

$$\Omega_{ar} = \frac{[Ca^{2+}][CO_3^{2-}]}{K_{sp}} \quad (1)$$





With decreasing carbonate ion concentration (e.g. as a result of surface ocean acidification), the saturation state is reduced, which can lead to decreased coral reef calcification (Chan and Connolly, 2013; Albright et al., 2018). It also leads to higher

dissolution, which could lead to coral reefs becoming net dissolving when atmospheric $CO_2$ reaches 560 ppm (Silverman et al., 2009), which is expected to occur by ~2050 (Eyre et al., 2018).

Numerous studies have investigated the impact of warming and/or acidification on coral reef organisms, either in situ (e.g. Albright et al., 2016, 2018; Sully et al., 2019) or in mesocosms/laboratories (e.g. Dove et al., 2013). Based on such studies,

numerical and statistical models have been developed and used to evaluate the impact of temperature and ocean acidification changes on coral reefs, often regionally (Evenhuis et al., 2015; Buddemeier et al., 2008; 2011; Sully et al., 2022), but also globally (Kleypas et al., 1999; Donner et al., 2005; Silverman et al., 2009; Frieler et al., 2012; Couce et al., 2013; van Hooidonk et al., 2014; Eyre et al., 2018; Cornwall et al., 2021). Among the models used to evaluate the impact on coral reefs during the next century globally, some considered the impact of future temperature change (Donner et al., 2005), others the impact of $\Omega_{ar}$

change (Kleypas et al., 1999, Eyre et al., 2018), and some both variables simultaneously (Silverman et al., 2009; Frieler et al., 2012; Couce et al., 2013; van Hooidonk et al., 2016; Cornwall et al., 2021; Cornwall et al., 2023). These coral reef models were not embedded within climate models; hence they were forced by climate data outputs, that are often only stored at monthly resolution. This means that daily temperature could not always be used, preventing accurate bleaching effect computation. In addition, the aragonite saturation state ($\Omega_{ar}$) is also often not stored, further hindering accounting for this effect. Here we use

a coupled numerical climate-carbon cycle model that includes a coral reef module (Bouttes et al., 2024), allowing us to evaluate the dynamics of coral reef carbonate production as a function of temperature and $\Omega_{ar}$ on daily timescales. The use of this coupled model allows us to evaluate the individual and combined impact of ocean warming and acidification, to test possible non-linearities.

In addition, future coral carbonate production is difficult to project due to several sources of uncertainties. (i) Different emission scenarios will yield different evolution pathways of climate and ocean acidification (Kwiatkowski et al., 2020). Past studies have tested the impact of different scenarios (Cornwall et al., 2021; 2023), but they were restrained to the old RCP scenarios. Here we span the different SSP scenarios (Meinshausen et al., 2020) that are currently used as inputs for state-of-the-art GCMs. (ii) Different climate models, that have various climate sensitivities (Forster et al., 2020; Meehl et al., 2020;

Forster et al., 2021), result in different climate and ocean acidification evolutions for the same emission scenario. To account for this uncertainty, we use different versions of the iLOVECLIM climate model spanning the range of climate sensitivities in climate models. (iii) Potential coral adaptation to thermal bleaching could also result in different coral reef carbonate production (Cornwall et al., 2023). Hence, we use two versions of the coral reef model: with or without adaptation. We make use of the fast coral-carbon-climate model to evaluate the impact of all three sources of uncertainties in future projections.





## 2 Methods

We use iLOVECLIM, a coupled climate-carbon cycle model which includes a new module of calcium carbonate production by coral reefs called iCORAL (Bouttes et al., 2023).

### 2.1 The iLOVECLIM-iCORAL climate model with coral reefs

### 2.2.1 iLOVECLIM carbon-climate model

iLOVECLIM is an intermediate complexity model including an atmosphere (ECBILT), an ocean (CLIO), as well as sea ice (LIM) and a continental vegetation module (VECODE) inherited from the LOVECLIM model (Goosse et al., 2010). The ocean model has a horizontal resolution of 3° with 20 vertical levels; the atmosphere has a horizontal resolution of 5.6° with 3 levels. It is also coupled to an ocean carbon cycle module (Bouttes et al., 2015). The ocean carbon cycle module includes two types of plankton (phytoplankton and zooplankton), labile and refractory dissolved organic carbon (respectively DOC and DOCs), dissolved inorganic carbon (DIC), alkalinity (ALK), and nutrients (PO4). The total particulate production that gets exported from the euphotic zone is progressively remineralised below. Phytoplankton, zooplankton, DOC, DOCs, DIC, ALK and nutrients are all transported (by advection-diffusion) in the ocean. The iLOVECLIM model is relatively fast with around 700 simulated years per day, making it well suited for both long duration and large ensemble simulations.

### 2.2.2 iCORAL coral reef carbonate production module

A coral module has recently been added in iLOVECLIM (iCORAL, Bouttes et al., 2024). It is based on the ReefHab model (Kleypas, 1995; Kleypas, 1997) with some modifications and add-ons. In each grid cell of the model, the module computes habitability, i.e. the possibility of coral reef development depending on temperature, salinity, phosphate concentration, and light limitation.

In habitable zones, coral reef carbonate production is computed on a vertical subgrid with a 1 m resolution from the available photosynthetically active radiation ($PAR$), temperature, $\Omega_{ar}$, surface area and topography. The module assumes the possibility of having coral development as soon as the conditions are favorable, without considering larvae dispersion. In developments to ReefHab, we have added temperature and $\Omega_{ar}$ as variables used to compute coral reef carbonate production. The impact of bleaching on carbonate production is also accounted for with options for coral reef recovery times following bleaching events of differing intensity (see Bouttes et al. (2024) for details). In iCORAL we consider only net carbonate production, i.e. there is no explicit computation of gross production nor dissolution. In the current study, the impact of coral reef carbonate production on the global carbon cycle is not considered, as its effect is likely to be small on centennial timescales.



A previous study comparing model results with existing data of coral reef carbonate production and location has shown
120    comparatively good agreement (Bouttes et al., 2024). The global mean carbonate production simulated in the model is 0.81 Pg
CaCO$_3$ yr$^{-1}$, which lies in the range of data estimates from 0.65 to 0.83 Pg CaCO$_3$ yr$^{-1}$ (Vecsei, 2004). The model simulates a
global coral reef area of 390×10$^3$ km$^2$, also in the range of observations, although these have large uncertainty, ranging from
150 ×10$^3$ km$^2$ to 1500 ×10$^3$ km$^2$ (Smith, 1978; Crossland et al., 1991; Copper, 1994, Spalding et al., 2001; Vecsei, 2004; Li
et al., 2020). The model correctly simulates the presence of coral reef in most locations where they are observed, but also in a
few places where there are no observations to date (Bouttes et al., 2024).

## 2.2 Simulations

We have run an ensemble of 31 simulations under historical and future greenhouse gas concentration scenarios, testing
different hypotheses with regard to climate model climate sensitivity, coral reef thermal bleaching adaptation and
socioeconomic scenarios. All simulations were run from 1850 (starting from an equilibrated 1850 run) to 2300 with prescribed
atmospheric CO$_2$, CH$_4$ and N$_2$O concentrations for the historical period and the different shared socioeconomic pathways
(SSPs; Meinshausen et al., 2020). These concentrations are publicly available and were downloaded from the Earth System
Grid Federation (ESG, https://esgf-node.llnl.gov/search/input4mips/). Changes in land use and concentrations of atmospheric
aerosols are not considered in the iLOVECLIM simulations.


### 2.2.1 Equilibrium climate sensitivity (ECS)

The climate response of models is strongly dependent on their equilibrium climate sensitivity (ECS). The ECS is the
equilibrium response of global mean temperature change to a doubling of atmospheric CO$_2$ concentration compared to pre-
industrial (i.e. from ~280ppm to 560ppm) and is often computed by regressing the top of atmosphere radiative flux against the
global average surface air temperature change (Gregory et al., 2004). The ECS varies greatly among climate models, from
around 1.5°C up to 5.6°C in CMIP6 (Forster et al., 2020; Meehl et al., 2020). Based on several lines of evidence, the IPCC
AR6 very likely range for the ECS is 2°C to 5°C, and the likely range 2.5 to 4.0°C (Forster et al., 2021).



We test the impact of different climate sensitivities by spanning a range of possible equilibrium climate sensitivities within the

iLOVECLIM model. The standard iLOVECLIM ECS is at the lower range, with ~2°C (Fig. 1). We can, however, vary ECS

in the model by modifying the $\alpha$ parameter used in the long wave radiation code following Timm and Timmerman (2007):

$$LWR = \alpha \, a(\lambda, \varphi, p, t_{season}) log \left[ \frac{CO_2(t)}{CO_2(t0)} \right]$$

Where $CO_2(t0)$ = 356 ppm is the reference $CO_2$ concentration. The transfer coefficient $a$ is a function of longitude, latitude,

height, and season (Chou and Neelin, 1996).

In this study, the ECS is computed as the difference between the mean of the last 100 years of equilibrium simulation with

$2xCO_2$ and the pre-industrial simulation, for each $\alpha$ value. The $\alpha$ parameter is varied between 1 (standard simulation) and 3.

While the standard ECS with α=1 is 2°C, it increases to 5.4°C with α=3. With α=1.5, the ECS is 2.9°C, thus falling within the

AR6 likely range (Fig. 1). We have chosen this value to run additional simulations with different SSP scenarios and idealized

experiments to analyze the response of coral reefs.

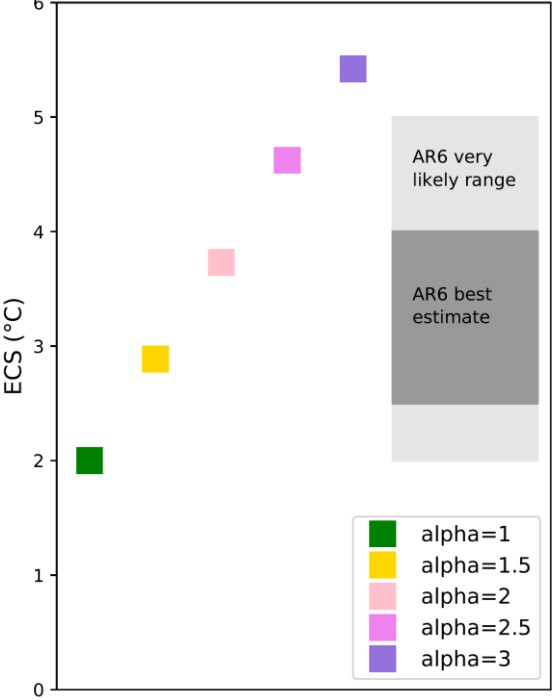

**Figure1: Equilibrium climate sensitivity (ECS, °C) in iLOVECLIM computed after 2000 years of simulation compared to the estimated range from IPCC AR6 (Forster et al., 2021).**



### 2.2.2 Coral thermal adaptation to bleaching

The potential adaptation of coral reefs to thermal conditions that can cause bleaching is poorly constrained (Cooley et al., 2022). We thus consider two contrasting cases of coral adaptation to bleaching: one with no adaptation and one with continuous adaptation.

In the coral reef module, the bleaching of corals is simulated when the cumulative temperature anomaly passes a threshold following NOAA's degree heating weeks (DHW) approach

(https://www.coralreefwatch.noaa.gov/product/5km/methodology.php#dhw). In each grid cell, the temperature anomaly is computed with respect to a reference, which is the maximum of the climatological monthly mean temperature over 30 years, i.e., the temperature of the hottest month in the climatological monthly means for each grid element (Maximum of the climatological Monthly Mean temperature, *MMM*). This is consistent with the approach of NOAA when providing coral bleaching hotspot predictions (https://www.coralreefwatch.noaa.gov/product/5km/methodology.php). In the simulations, we

evaluate the impact of possible coral adaptation on the thermal threshold for bleaching by changing the reference period used to calculate the MMM. Assuming non-adaptation, this reference is computed from the first 30 years of the simulations (hence from 1850 to 1879) and kept fixed. On the contrary, if we assume adaptation to changing temperature (referred to as "thermal adaptation to bleaching" in the following), this temperature reference evolves with time and is computed from the 30-year running mean.


### 2.2.3 Greenhouse gas scenarios

We run each simulation starting from 1850 under the historical scenario followed by different SSPs (Fig. 2). Depending on the simulation, we consider the SSP1-1.9, SSP1-2.6, SSP2-4.5, SSP3-7.0, SSP5-3.4-OS or SSP5-8.5 (O'Neill et al., 2016; Meinshausen et al., 2020). For each ECS, with- and without thermal adaptation to bleaching, we run simulations with the low-

emission scenario SSP1-2.6 and the high-emission scenario SSP5-8.5, to bound the projected response of coral reef habitability





and carbonate production (Table 1). In the case of the mid-range ECS of 2.9°C (for α=1.5) we have additionally run simulations

with all SSPs to evaluate more precisely the impact of the different scenarios (Table 1).

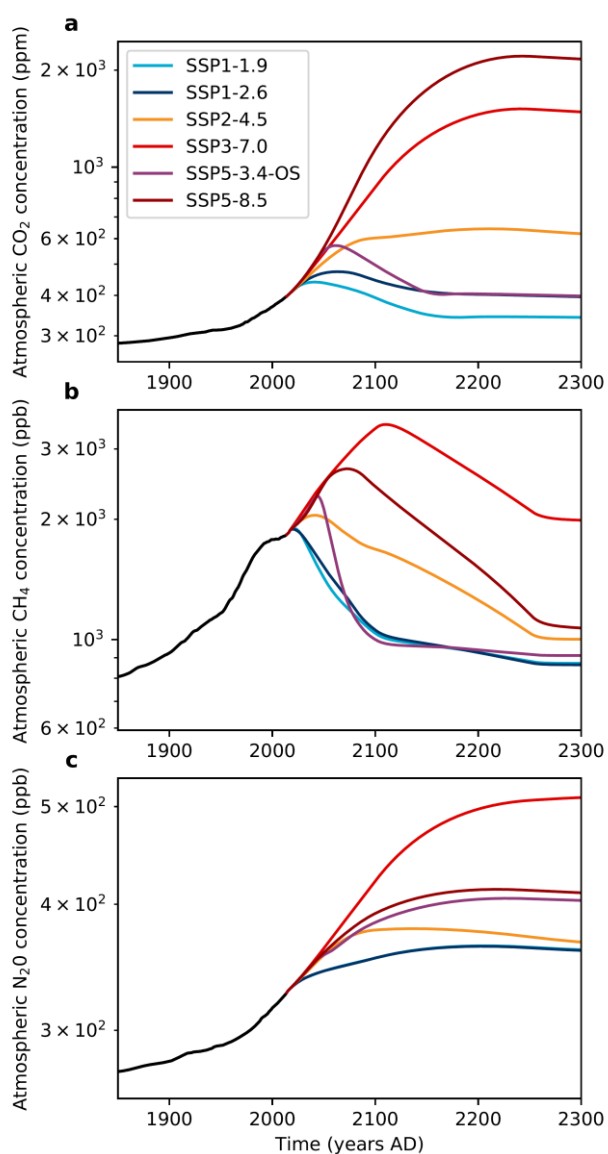

**Figure 2: Evolution of atmospheric CO₂, CH₄ and N₂O for the different SSP scenarios (data from Meinshausen et al., 2020).**



| α<br>Scenario | 1.0 | 1.5 | 2.0 | 2.5 | 3.0 |
|---|---|---|---|---|---|
| SSP1-1.9 | | + | | | |
| SSP1-2.6 | + | + | + | + | + |
| SSP2-4.5 | | + | | | |
| SSP3-7.0 | | + | | | |
| SSP5-3.4-OS | | + | | | |
| SSP5-8.5 | + | +<br>$\Omega T$, $T$, $\Omega$ | + | + | + |

**Table 1: Overview of the climate scenario–model climate sensitivity combinations used for the simulation experiments. All simulations marked by a plus sign (+) were carried out in two variants: one with coral thermal adaptation to bleaching (moving 30-year reference climatology), and one without (reference climatology fixed to first 30 years of the simulation). For the SSP5-8.5 scenario, three additional simulations were carried out to allow for a simple factor impact analysis: "$\Omega T$" for which the $T$ and $\Omega_{ar}$ distributions seen by the corals are kept fixed at the values prevailing at the end of the first simulation year; "T" for which only the temperature distribution is kept fixed, but the $\Omega_{ar}$ distribution evolves as calculated by the carbon cycle model; "$\Omega$" for which the $\Omega_{ar}$ is kept fixed, but T follows the evolution calculated by the climate model. For these three, bleaching adaptation was disabled.**

### 2.2.4 Relative roles of warming and acidification

Finally, we have also run idealized SSP5-8.5 simulations with an ECS of 2.9°C (α=1.5) where coral reef habitability and carbonate production are considered to be (i) independent of temperature (including bleaching), (ii) independent of $\Omega_{ar}$, and (iii) independent of both temperature (including bleaching) and $\Omega_{ar}$ (Table 1). The temperature and $\Omega_{ar}$ that coral reefs experience in these idealized simulations is set to those reached at the end of the first simulated year (i.e. 1850), removing the impact of warming and acidification on carbonate production. These simulations allow analysis of the individual impacts of warming, acidification and bleaching on coral reef habitability and carbonate production, allowing assessment of the relative importance of these stressors and potential non-linearities when there is compound stressor exposure.





## 3 Results

### 3.1 Temperature and pH

The main environmental variables controlling coral reef carbonate production in the model are temperature and $\Omega_{ar}$. While temperature is available from CMIP6 models, $\Omega_{ar}$ is not a standard output, so that we compare results for pH which is closely related to $\Omega_{ar}$. The sea surface temperatures in iLOVECLIM are similar to the CMIP6 temperatures in terms of time evolution (Fig3 a and b). However, the amplitude of the warming is smaller in iLOVECLIM compared to CMIP6. Even with high ECS,

the warming in iLOVECLIM is in the low range of CMIP6 projections. Hence the resulting coral reef production simulated by the model might be conservative in the sense that its decline might be underestimated. The evolution of the pH is very similar in iLOVECLIM and CMIP6. It does not depend on the ECS as it is mainly driven by the atmospheric $CO_2$ concentration.

**Figure 3: Evolution of global sea surface temperature and pH for CMIP6 models compared to the iLOVECLIM model for SSP1-2.6**
**and SSP8-5.8. The variables are anomalies with respect to the mean value for 1870-1899. The CMIP6 data are from Kwiatkowski et al., 2020.**



## 3.2 Projected coral reef distribution and carbonate production

During the historical period, the coral production decreases very slowly, from 0.81 Pg CaCO$_3$ yr$^{-1}$ in 1850 to 0.74 Pg CaCO$_3$

yr$^{-1}$ in 1970 (Fig. 4). At the end of the 20$^{th}$ century, the production starts decreasing more strongly in all simulations, but the

evolution diverges depending on scenarios and the possibility of adaptation.

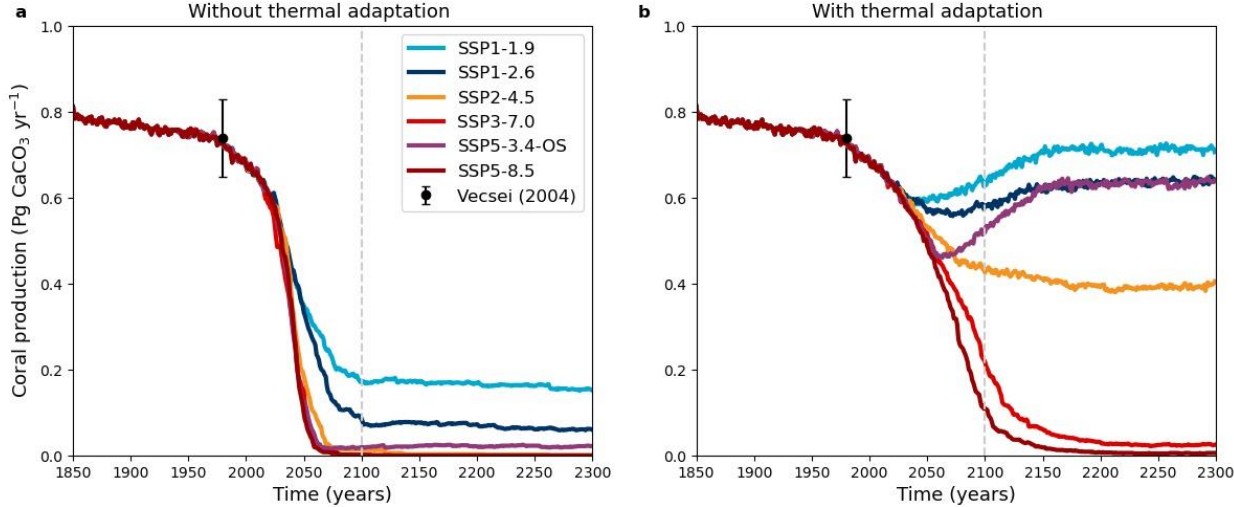

**Figure 4: Dynamics of the global coral reef CaCO$_3$ production (Pg CaCO$_3$ yr$^{-1}$) over the historical simulation and for different SSP scenarios, (a) without and (b) with thermal adaptation to bleaching. The iLOVECLIM version used for these simulations has an**

**ECS of 2.9°C ($\alpha$ =1.5, Fig. 1). The circle with whiskers indicates estimates of the modern data of CaCO$_3$ production (Vecsei, 2004).**

## 3.2.1 Without thermal adaptation to bleaching

In the iLOVECLIM simulations without bleaching adaptation, that have an ECS of 2.9°C, coral carbonate production rapidly

declines under all SSPs in the early 21$^{st}$ century (Fig. 4, left panel). The rate of this decline is however lower with high-

mitigation scenarios such as SSP1-1.9 and SSP1-2.6. By 2100, coral reef carbonate production either ceases entirely (SSP 2-

4.5, SSP3-7.0, SSP5-8.5) or stabilizes at 25% or less of the preindustrial value (SSP1-1.9, SSP1-2.6). With the overshoot

scenario (SSP5-3.4-OS) carbonate production stops then slightly recovers, albeit at a very low level.




### 3.2.2 With thermal adaptation for bleaching

Bleaching adaptation strongly modifies the results of our projections of coral reef carbonate production over the next three centuries (Fig. 4, right panel). Across all scenarios, the rate of carbonate production decline is reduced, with some carbonate production still occurring in all scenarios in 2100 (contrary to the simulations without adaptation).

Under the high-emission scenarios SSP3-7.0 and SSP5-8.5, coral reef carbonate production still eventually ceases (or becomes negligibly small in the case of SSP3-7.0), albeit ~140 years later than when no bleaching adaptation is assumed (~2060 without adaptation vs. ~2200 with adaptation). By 2100, carbonate production is still considerably diminished with SSP5-8.5, with the location of net accreting coral reefs reduced in all oceans (Fig. 5d) compared to the pre-industrial (Fig. 5a). Under SSP2-4.5 (medium emissions) reef carbonate production decreases and stabilizes at around half the pre-industrial value and by 2100

most coral reef locations are still net accreting (Fig. 5c).

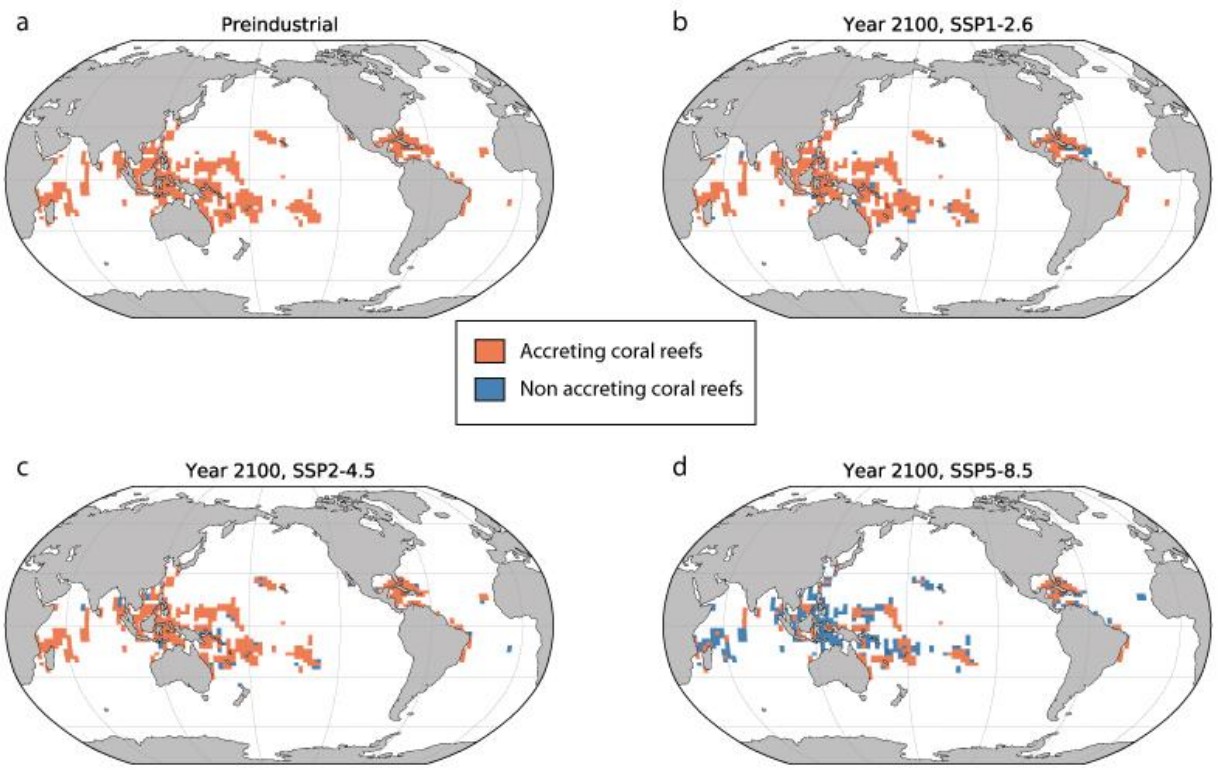



**Figure 5: Simulated distributions of net accreting coral reefs under (a) pre-industrial conditions, and in 2100 (mean of 2095-2105) under (b) SSP1-2.6, (c) SSP2-4.5 and (d) SSP5-8.5. An ECS of 2.9°C (α =1.5) was adopted for all simulations, and thermal adaptation** 250 **to bleaching was activated.**

Carbonate production initially decreases and then partially recovers under the high-mitigation scenarios (SSP1-1.9 and SSP1-2.6) as well as under the overshoot scenario (SSP5-3.4-OS), with most coral reef locations returning to net accreting in 2100

(Fig. 5b). With the overshoot scenario, regions where coral reefs were not accreting by ~2050 can become net accreting again in the following years as coral production recovers (Fig. 6).

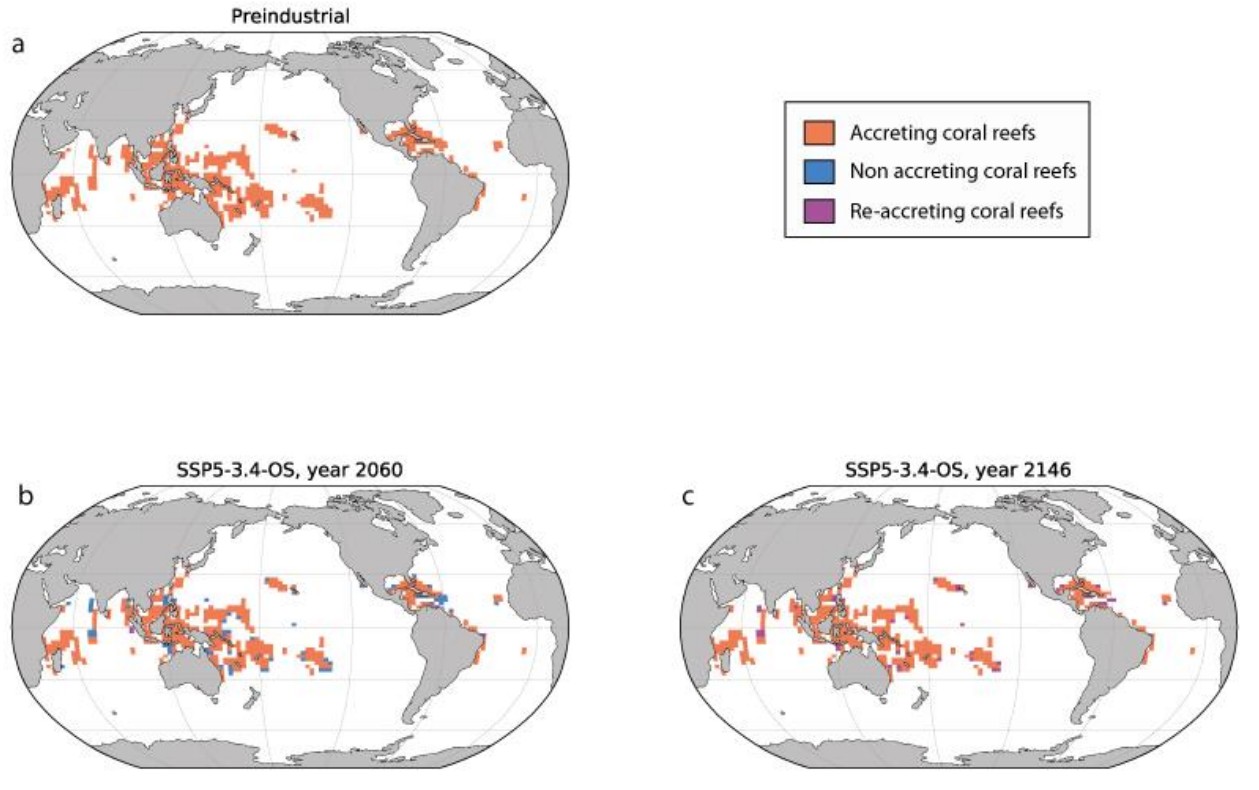

**Figure 6: Simulated distributions of net accreting coral reefs under (a) pre-industrial conditions, and in SSP3.4-OS at (b) year 2060** 260 **(compared to pre-industrial), (c) year 2146 (compared to year 2060). An ECS of 2.9°C (α =1.5) was used for this simulation, and thermal adaptation to bleaching was activated.**





At the end of the 21$^{st}$ century under the high-mitigation scenario SSP1-2.6, global reef carbonate production is 10% of the pre-industrial value if there is no bleaching adaptation compared to 70% with adaptation. With SSP5-8.5, carbonate production

ceases entirely without adaptation but is reduced to 10% of the pre-industrial with adaptation (Fig. 7).

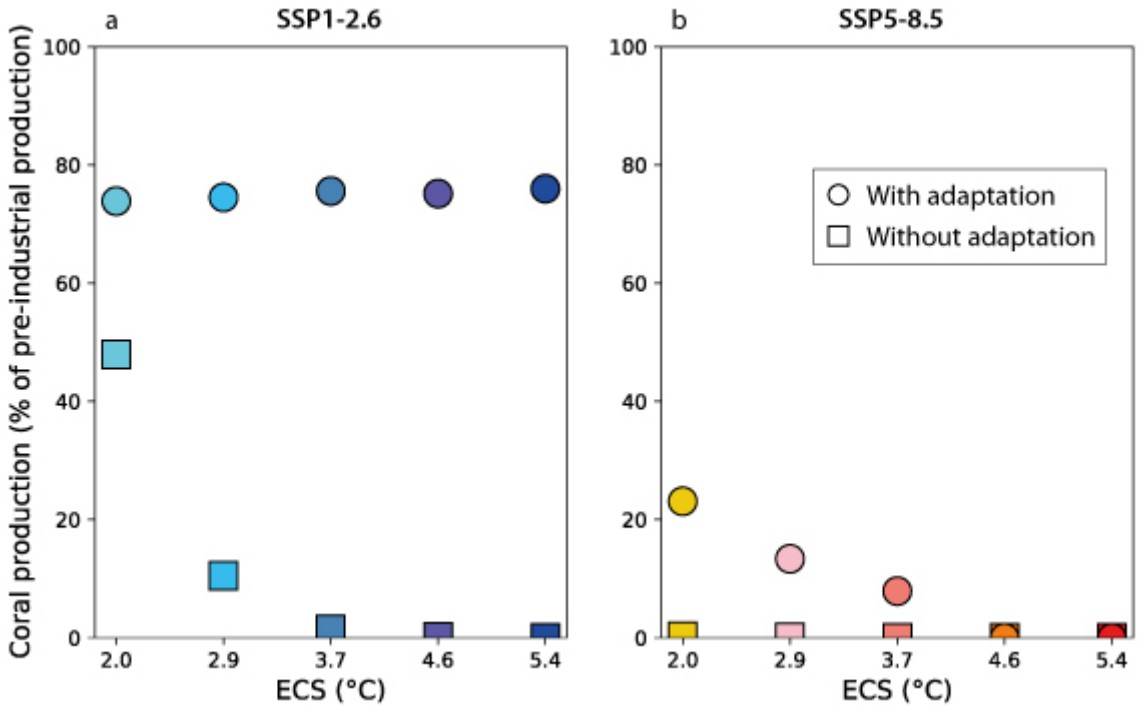

**Figure 7: Percentage of pre-industrial global coral reef carbonate production simulated in the year 2100 for SSP1-2.6 and SSP5-8.5 across ECS values (percentage values are 2090-2110 means relative to 1850-1870 mean carbonate production).**


### 3.3 Impact of equilibrium climate sensitivity

Across the range of ECS values, the large difference in the projected carbonate production between high-mitigation (SSP1-2.6) and high-emission (SSP5-8.5) simulations is maintained when adaptation is considered (circles on Fig. 7). With SSP1-2.6, the temperature and saturation state changes are much smaller than with SSP5-8.5 (Fig. 3). Accordingly, the global

carbonate production decrease is smaller with SSP1-2.6 than with SSP5-8.5, as both increased temperature and decreased saturation state tend to lower production. In addition, when we let the reference temperature climatology for bleaching evolve

with time, coral reefs have time to adapt to such temperature changes and will be less prone to bleaching. By 2100, this leads to a relatively small decrease of carbonate production of less than 30% (production is maintained at levels above 70% of the pre-industrial value) with SSP1-2.6. On the contrary, with SSP5-8.5 the temperature and $\Omega_{ar}$ changes are much larger, and the

temperature change is too fast for corals to adapt sufficiently to avoid bleaching. This results in a greater than 75 % decrease in carbonate production by 2100 (<25% of pre-industrial carbonate production).

If no adaptation is considered (squares on Fig. 7), with SSP1-2.6 carbonate production in the simulations decreases with increasing ECS, as there is greater ocean warming. By 2100, it has decreased to levels between 0 to 50 % of the pre-industrial production. With SSP5-8.5, ocean warming is so strong that carbonate production ceases before the end of the 21$^{st}$ century for

all ECS values.

### 3.4 Long term (after 2100) changes in carbonate production

After 2100, with SSP1-2.6 carbonate production partially recovers when thermal adaptation is considered, so that it is higher in 2300 (at ~80%) compared to 2100 (at ~70%), but still lower than the pre-industrial level (circles on Fig. 8, left panel

compared to Fig. 7, left panel). Without adaptation, the carbonate production remains low or ceases (squares on Fig. 8, left panel). With SSP5-8.5, the coral reef carbonate production drops to zero, when it has not already done so before, in all simulations even when thermal adaptation to bleaching is considered (Fig. 8, right panel).




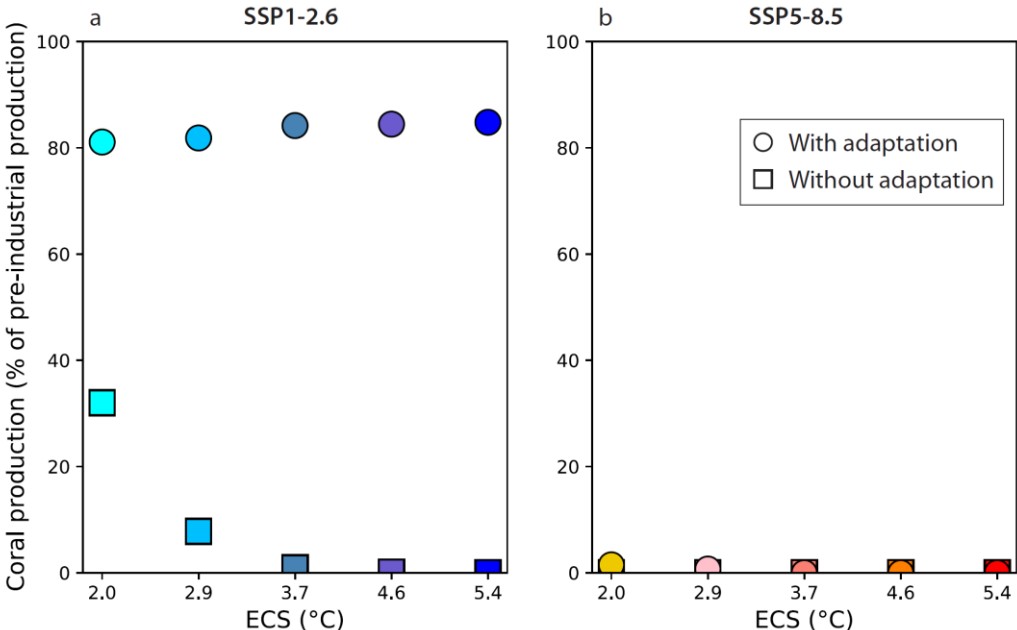

**Figure 8: Percentage of pre-industrial global coral reef carbonate production simulated in year 2300 for (a) SSP1-2.6 and (b) SSP5-8.5 across ECS values (percentage values are 2280-2300 means relative to 1850-1870 mean carbonate production).**

## 3.5 Relative influence of warming and acidification

To evaluate the relative role of saturation state and temperature (including bleaching) we have run idealized simulations by

removing separately (or in combination) their limiting effects on carbonate production (Fig. 9). Having neither temperature

nor $\Omega_{ar}$ limitation (blue line, 'other drivers') results in a persistent coral reef carbonate production, showing that future changes

of other variables such as nutrients or light do not significantly influence coral production. The reduction in $\Omega_{ar}$ suppresses

simulated carbonate production. When only $\Omega_{ar}$ limitation is accounted for in coral production (effect of acidification, no

temperature limitation considered, green line) carbonate production initially responds similarly to the standard simulations

with all limiting variables (red line). However, around the year 2030 the temperature starts to play a larger limiting role (effect

of warming, no $\Omega_{ar}$ limitation, orange line). Hence both temperature (including bleaching) and $\Omega_{ar}$ play a crucial role in

governing coral production, with $\Omega_{ar}$ being the main limiting factor in the early part of the simulation, while temperature

becomes the main limiting variable later. The effects of warming and acidification combine linearly during the first part of the

evolution until around year 2040, corresponding to slightly less than 50% of the production decline (purple line compared to



red line). The response of the production then becomes non-linear, with a saturation of the production decline which tends to

mainly follow the effect of warming.

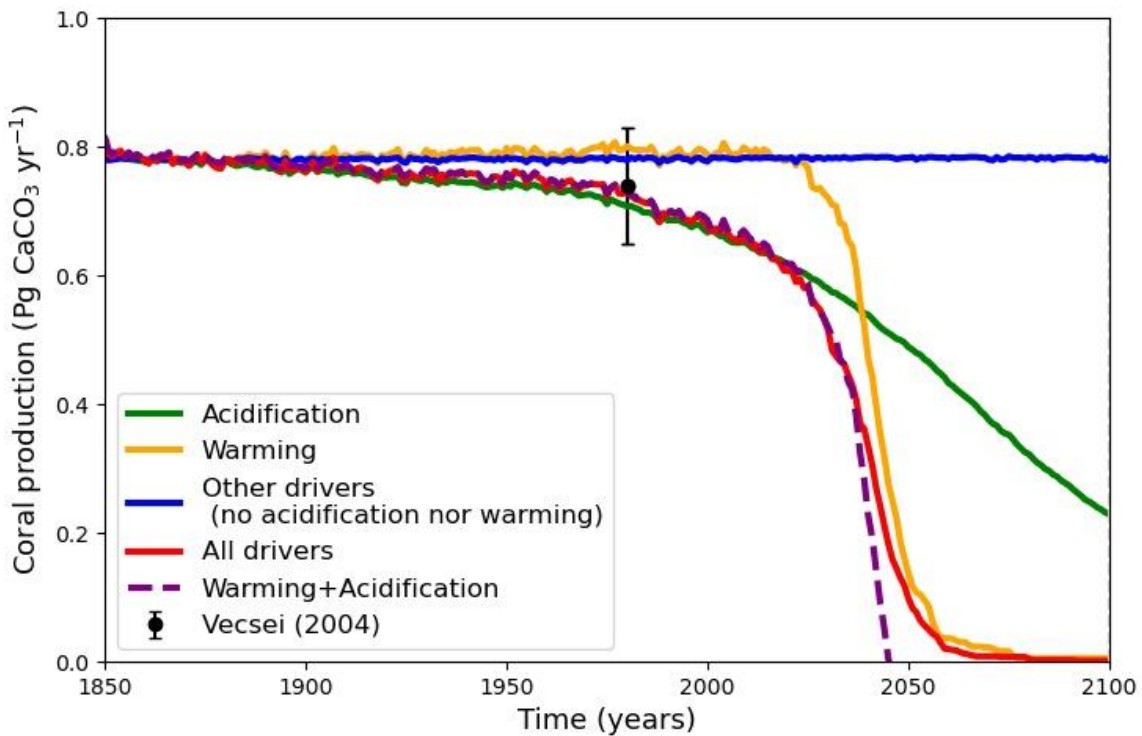

**Figure 9: The projected impact of acidification, warming and other drivers on global coral reef carbonate production (Pg CaCO₃**
**yr⁻¹) for ECS= 2.9°C (α=1.5) for the historical+SSP5-8.5 scenarios. Simulations are without thermal adaptation to bleaching. The**
**purple dashed line is the linear combination of the warming and acidification effects.**

## 4 Discussion

### 4.1 Comparison with past studies

While most previous studies focused on specific regions, or only accounted for the individual impact of warming or

acidification, Cornwall et al. (2021; 2023) quantified future global changes in coral reef calcification in response to changes

in both temperature and $\Omega_{ar}$. They used a different method, based on statistical functions derived from observations. For the

future, they used model outputs from simulations forced by representative concentration pathway (RCP) scenarios. Although





the new SSP scenarios are different, RCP2.6 and SSP1-2.6, as well as RCP8.5 and SSP5-8.5 are comparable (Meinshausen et

al., 2020). By construction, the SSP and corresponding RCP scenarios have the same radiative forcing in 2100 (e.g. 8.5 W m⁻² for RCP8.5 and SSP5-8.5. However, as discussed in Kwiatkowski (2020) the greenhouse gas emissions, hence concentrations, differ due to different mixes of energy sources. For example, the $CO_2$ concentration is higher under SSP5-8.5 compared to RCP8.5, which results in greater ocean acidification in SSP5-8.5 compared to RCP8.5.

By 2100, Cornwall et al. (2021) projected a mean decline in global net carbonate production of 77% for RCP2.6 and complete cessation for RCP8.5. With our coupled climate-coral model we also project complete cessation of net accretion by 2100 for SSP5-8.5 (in the absence of thermal adaptation), but the results for SSP1-2.6 strongly depend on the model ECS. It varies from a 50% reduction with a very low ECS to complete cessation with high ECS. Cornwall et al. (2023) additionally considered possible coral adaptation to warming (symbiont evolution and symbiont shuffling) based on ecological modelling (Logan et al., 2021). They showed that the main factor driving carbonate production was the emission scenario, with adaptation changing the severity of the impacts. Adaption only had a significant effect in the low-emission scenario (RCP2.6) in this study. We also note a larger impact of thermal adaptation with the low-emission scenario. However, with low ECS thermal adaptation also modifies the resulting carbonate production. In our simulations, thermal adaptation is not considered through ecological parameterizations such as in Logan et al. (2021), which would be too complex relative to the rest of the model. We consider thermal adaptation through changes in the reference temperature for the bleaching scheme. With adaptation, the time window used to compute the temperature reference (the maximum of the climatological monthly mean temperature over 30 years) evolves, while it is kept constant (at the beginning of the simulation) otherwise. This thermal adaptation strongly modifies the results in our simulations, especially for low emission scenarios as SSP1-2.6 where the carbonate production decline is substantially reduced compared to the simulations without thermal adaptation, and maintained above 70% of pre-industrial levels for SSP1-2.6 across all ECS values. Hence, we obtain similar values, especially without thermal adaptation, but we show that accounting for different climate sensitivities, which would correspond to uncertainties among climate models, also results in large differences. In addition, thermal adaptation strongly modifies the resulting carbonate production, in particular with high-emission scenarios, but also with low-emission scenarios when combined with low ECS.



## 4.2 Caveats and missing processes

Because we have embedded a coral reef carbonate production module within a climate model, the results depend on the climate

simulated by the model. In iLOVECLIM, the temperature change tends to be at the lower end of the range covered by CMIP6

simulations. Hence, simulated carbonate production reductions might be underestimated. iLOVECLIM has a relatively coarse

ocean resolution of 3° by 3° on the horizontal grid. While we have been able to take into account the spatial heterogeneity of

the seafloor topography by adopting a subgrid parametrization, this is not possible for other variables such as temperature, as

they depend on local circulation dynamics. Hence the model cannot account for small scale features such as local temperature

and $\Omega_{ar}$ changes. A consequence of this is that while large-scale changes can be evaluated with the model, local changes

dependent on small-scale dynamics cannot be simulated. Such limitation would be alleviated by the use of a higher resolution

model in which the same coral module could be implemented. In addition, higher resolution models also have a higher temporal

resolution, resulting in better tropical variability, which would further improve the modelled coral reef response.

The coral reef carbonate production module strongly relies on current knowledge of the coral reef response to environmental

variables. Large uncertainties remain and better understanding of coral reef responses will help improve the coral module in

the future. For example, the relationship between $\Omega_{ar}$ and the rate of calcification is complex (Chan and Connolly, 2013; Jokiel,

2016; Eyre et al., 2018) and further research is required to refine the model parameterizations. In addition, we have shown that

the possibility of adaptation to thermal stress results in a wide range of responses. More constraints on the potential for

adaptation (e.g. Logan et al., 2021) will help to reduce the range of future coral reef carbonate production projections.

## 4.3 Perspectives

A current limitation in the coral reef module is the use of a common function to represent the response of all coral reefs to

environmental conditions. Coral species respond differently to changes in temperature and $\Omega_{ar}$ (Klepac et al., 2023) and species

composition differs across reefs. In addition, the different components of coral reefs (coral, algae, crustose coralline algae,

sediment) also substantially differ across reefs and each have different sensitivities to temperature and saturation state changes

(Kroeker et al., 2010; Eyre et al., 2018; Leung et al., 2022). A future improvement could thus be to consider several coral



functional types, allowing for example to build reefs dominated by branched vs massive corals, similarly to what is done in land vegetation models with plant functional types.


In the coral reef module, we compute net carbonate production, which represents the net difference between production and destruction (erosion and dissolution) in an implicit manner. However, we do not explicitly simulate the impact of bioeroders, such as sponges, or crown-of-thorns starfish. As they play an important role in counteracting carbonate production (Schonberg et al., 2017), accounting for them could improve the model, but this would require a complex ecological model.

Finally, we do not consider other anthropogenic impacts on corals, such as pollution or overfishing, which can further degrade living conditions for coral reefs, nor local and regional management, which can offset some of that stress (Wolff et al., 2018).

## 5 Conclusions

Using a global coupled coral-climate model, we have shown diverse carbonate production changes from coral reefs in future

projections. The large range of carbonate production changes stems from uncertainties in scenarios (socio-economic uncertainties), climate sensitivities (climate model uncertainties) and thermal adaptation (coral reef biology uncertainties). With the high-emission SSP5-8.5 scenario, global coral reef carbonate production drops dramatically or ceases, regardless of the equilibrium climate sensitivity (ECS) and potential thermal adaptation to bleaching. Only with a low ECS and thermal adaptation to bleaching can coral reef carbonate production be kept stable under SSP5-8.5, albeit at a fraction of the pre-

industrial value (less than 25% of the preindustrial value with ECS=2°C in 2100). On the contrary, with the low-emission SSP1-2.6 scenario, the potential range of future coral reef carbonate production is much larger and can reach 76% of the preindustrial value in 2100 (with ECS=5.4°C and thermal adaptation). Carbonate production depends strongly on the possibility of thermal adaptation. For SSP1-2.6 without thermal adaptation, global carbonate production drops to between 0% and 48% of pre-industrial values in 2100 and between 0% and 32% of preindustrial values in 2300. With thermal adaptation,

the production decreases are much more moderate, the carbonate production drops to between 73% and 76% of pre-industrial values by 2100, recovering to between 81% and 85% by 2300. With the SSP5-3.4 overshoot scenario, the conditions can



become habitable for coral reefs to become net carbonate accretors again in some regions, but this assumes that larvae are available to repopulate these regions.


**Code availability**

The code of the iCORAL module is available on Zenodo (https://doi.org/10.5281/zenodo.7985881; Bouttes et al., 2023).

**Data availability**

The data that support the findings of this study are openly available in Zenodo at http://doi.org/10.5281/zenodo.12958336.

**Author contribution**

Nathaelle Bouttes: Conceptualization; funding acquisition; investigation; methodology; writing – original draft preparation; writing – review and editing

L. Kwiatkowski: Conceptualization; methodology; writing – review and editing

E. Bougeot: investigation

M. Berger: investigation

V. Brovkin: Conceptualization; methodology; writing – review and editing

G. Munhoven: Conceptualization; methodology; writing – review and editing


**Competing interests**

The authors declare that they have no conflict of interest.

**Acknowledgments**

We thank Didier Roche for his support with the iLOVECLIM model. Financial support for this work was provided by the Belgian Fund for Scientific Research – F.R.S.-FNRS (project SERENATA, grant no. CDR J.0123.19). Guy Munhoven is a Research Associate with the Belgian Fund for Scientific Research – F.R.S.-FNRS. Victor Brovkin acknowledges funding by the European Research Council (ERC) as part of the Q-Arctic project (grant agreement No 951288).  Lester Kwiatkowski



acknowledges funding by the European Research Council (ERC) as part of the TipESM project (grant agreement No
101137673) and the ENS Chanel research chair. The authors acknowledge the ANR – FRANCE (French National Research
Agency) for its financial support of the TICMY project n°273305.

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
