# Peer review of "Projections of coral reef carbonate production from a global climatecoral reef coupled model"

_EGUsphere, 2024_

## Author Comment (AC1)

**Reply on Reviewer 1**

The mansucript attempts to project global changes in carbonate production using a habitat suitability model coupled with a climate model. The methods regarding how carbonate production is calculated, and the exact methods forcing changes in net carbonate production under climate change need to be made explicit here in the methods. Currently, the reader must go through a number of other papers to determine why the results of this mansucript have played out how they have and what the authors have actually done. I also found the writing to not be very direct. Clarifying points that are trying to be made, and explaining clearly why, would make this mansucript more accessible to a general reader. This mansucript needs to be revised before its suitably could be assessed, but likely would provide the reader a useful alternative to other existing models of changes in carbonate production under climate change.

We thank the reviewer for their comments. As suggested we have tried to add more information so that it is easier to understand without reading the other papers.

I give specific comments below that hopefully will assist the authors:

Line 39 onwards: What does the plus/minus indicate? Standard error, range, standard deviation? And is this the variability globally spatially or variability in any one location depending on the model outcomes?

The plus/minus indicates the standard deviation of the multi-model global change, more precisely the intermodel standard deviation, we have added precision in the new manuscript:

"Climate models indicate that under the high-emission scenario SSP5-8.5, sea surface temperatures will increase by 3.47 ± 0.78 °C, while the surface pH will decrease by -0.44 ± 0.005 by the end of the century (multi-model global mean change values of 2080–2099 relative to 1870–1899 ± the intermodel standard deviation, Kwiatkowski et al, 2020)."

Line 53: The role of carbonate ions in seawater has largely been disproven, see Comeau et al. (Comeau et al. 2018) and the various opinion/discussion papers, e.g., Cyronak et al (2015), Jokiel (2013).

The text here has been revised to reflect the general impact of acidification and not carbonate ions specifically. We agree that this part of the module could be improved in the future, as more data become available to derive a function of carbonate production from pH or DIC instead of saturation state. In the section on caveats and missing processes we have added more discussion:

"The calcification process which enables organisms to produce their external calcium carbonate skeleton requires more energy as the oceans take up anthropogenic carbon and acidify."

"For example, the relationship between $\Omega_{ar}$ and the rate of calcification is complex (Chan and Connolly, 2013; Jokiel, 2016; Eyre et al., 2018). As corals can control their internal pH value and seem to be more sensitive to pH, carbonate production might need to depend on pH instead of saturation state (Comeau et al., 2018)."

Line 69: And some used pH instead of saturation state, which is likely more appropriate for most calcifying taxa (including corals).

With have added pH in the manuscript:

"Among the models used to evaluate the impact on coral reefs during the next century globally, some considered the impact of future temperature change (Donner et al., 2005), others the impact of $\Omega_{ar}$ or pH change (Kleypas et al., 1999, Eyre et al., 2018), and some both variables simultaneously (Silverman et al., 2009; Frieler et al., 2012; Couce et al., 2013; van Hooidonk et al., 2016; Cornwall et al., 2021; Cornwall et al., 2023)."

Line 84: Please define GCM if it has not already been done so. But, does it need an acronym?

As it is not used elsewhere we agree there is no need for an acronym, it has thus been replaced by "Global Climate Models".

Line 86: Some intro to this model is required for the reader.

The description of the model is given in the method section: "2.2.1 iLOVECLIM carbon-climate model". We have indicated this in the manuscript: "To account for this uncertainty, we use different versions of the iLOVECLIM climate model (see section 2.2.1) spanning the range of climate sensitivities in climate models."

Line 120: There needs to be some more details here and further on regarding how this model is working for this to be a stand alone paper. Citing the previous paper and not including any details here means the reader must go through the Bouttes et al 2024 paper with a fine tooth comb to understand some very important aspects of this paper (probably the two most important parts of the methods) 1) how carbonate production is calculated and what controls it within the model, and 2) how temperature and ocean acidification impact carbonate production in this model. If space is an issue, remove the previous text that describes components of the model that are not as important for understanding the results here please.

As suggested we have added more details on the computation of carbonate production in the module:

"Carbonate production can take place provided that:

- The temperature is between 18.1°C and 31.5°C and exceeds 18.1°C throughout the year.
- The salinity is between 30 and 39
- The phosphate concentration is below 0.2 μmol L$^{-1}$
- The depth $Z$ is shallower than the maximum coral production depth ($Z_{max}$) which depends on attenuation of light in the water column:

$$Z_{max} = \frac{\log\left(\frac{I_{min}}{PAR}\right)}{K_{490}} \qquad (1)$$

where $I_{min}$ is a fixed parameter (the minimum light intensity necessary for reef growth), PAR is the photosynthetically active radiation at the surface (computed by the iLOVECLIM climate model) and $K_{490}$ is the diffuse attenuation coefficient at 490 nm taken from the Level-3 binned MODIS-Aqua products in the OceanColor database (available at: http://oceancolor.gsfc.nasa.gov). The production depth is defined as the depth at which light is at the Imin level.

In habitable zones, coral reef carbonate production $P$ is computed on a vertical subgrid with a 1 m resolution from the available photosynthetically active radiation ($PAR$), temperature $T$, aragonite saturation state $\Omega_{ar}$, surface area $S_{avail}$ and a topographic factor $TF$, following:

$$P = g_{max} \times f_R(PAR) \times f_T(T) \times f_O(\Omega_{ar}) \qquad (2)$$
$$\times S_{avail} \times TF$$
$$\times f_B(t; t_{bleach})$$

Where $g_{max}$ is the maximum value (fixed) and $f_B(t; t_{bleach})$ a function for the bleaching.

The two main potential drivers of production changes in the future are temperature and saturation state, as they will evolve in the future. In the coral module, the temperature function $f_T(T)$ is a linear function of temperature ($T$), °C, fitted for the temperature range of coral reef habitability ($f_T(T) = 0$ at T = 18.1°C and $f_T$ (T) = 1 at T = 31.5°C; $f_T$(T) = 0 outside the range of 18.1–31.5°C):

$$f(T) = -1.38 + 0.077 \times T \qquad (3)$$

Following Langdon and Atkinson (2005), the saturation state function $f_O(\Omega_{ar})$ is:

$$\text{if } \Omega > 1 \ f_O(\Omega) = \frac{\Omega - 1}{K_{omega}} \qquad (4)$$
$$\text{Else } f_O(\Omega) = 0$$

with $K_{omega}$ is a normalisation parameter ($K_{omega} = 2.86$)."

Paragraph around line 140: Please in clear language explain to the reader why this part of the methods is important? How does ESC matter in the context of coral reef carbonate production over absolute changes in temperature and pH? There is more details on this than on how carbonate production is estimated.

The ECS determines the simulated warming for a fixed amount of $CO_2$ emissions. Higher ECS leads to higher ocean warming, with large impacts on coral reefs. The ECS has very limited impact on ocean pH change as this is primarily driven by ocean carbon uptake, which is almost entirely determined by the concentration of atmospheric $CO_2$, with limited influence of simulated warming.

In the manuscript we have added:

"The simulated ocean warming in response to atmospheric $CO_2$ emissions and the resulting impact of this on coral reefs, is strongly dependent on the equilibrium climate sensitivity (ECS) of a given model."

Line 173: Its difficult to determine the speed and extent to which corals will gain increased tolerance to higher temperatures. However, this method is just as good as those previously used.

Paragraph around line 200: Other than using the NOAA guidelines for bleaching, and possible reef habitability, how do these projections of coral presence/absence actually function? A grid either has coral carbonate production at full value or 0 if the grid is habitable? Again, not enough detail on what are the most important aspects of the model outputs for this manuscript here.

The coral reefs are present if the net carbonate production is positive. Simulated carbonate production is dynamically simulated as a function of light, temperature, saturation state, surface area and topography and can vary between 0 and $g_{max}$. This is now described in detail in section 2.1.2 . The paragraph referred to by the reviewer here describes idealised simulations where some of the variables (temperature or saturation state) have prescribed fixed values in order to evaluate their respective impacts on carbonate production.

Line 330: Is global mean net erosion that same as complete cessation? If some locations still have positive net carbonate production, then perhaps this statement is misleading, as to me this means all locations stop producing carbonate.

Complete cessation means there is no net carbonate production (P=0) and is not the same as mean net erosion (P<0). Erosion is not explicitly accounted for in the model with negative P currently not permissible. We have modified the text here to avoid this confusion:

"By 2100, Cornwall et al. (2021) projected a decline in global mean net carbonate production of 77% for RCP2.6 with negative global net production (erosion) and no reefs able to accrete at rates equivalent to projected sea level rise for RCP8.5. With our coupled climate-coral model net erosion is not currently permissible, however we do project complete cessation of net accretion by 2100 for SSP5-8.5 (in the absence of thermal adaptation)."

Line 355: No model can project changes in in situ temperature on specific reefs at enough resolution anyway, so perhaps this does not matter.

Higher resolution models are capable to simulate temperature extremes closer to observations, so it could improve results. We plan to test this in the future.

References used here:

Comeau, S., C. E. Cornwall, T. M. DeCarlo, E. Krieger, and M. T. McCulloch. 2018. Similar controls on calcification under ocean acidification across unrelated coral reef taxa. Global Change Biology **24:** 4857-4868.

Cyronak, T., K. G. Schulz, and P. L. Jokiel. 2015. The Omega myth: what really drives lower calcification rates in an acidifying ocean. ICES Journal of Marine Science **73:** 558-562.

Jokiel, P. L. 2013. Coral reef calcification: carbonate, bicarbonate and proton flux under conditions of increasing ocean acidification. Proceedings of the Royal Society B: Biological Sciences **280:** 1764.

---

## Author Comment (AC2)

**Reply on Reviewer 2**

This is a high-quality study that adds to the literature on projecting coral reef futures with climate models over the remainder of this century. Although the results are not highly novel, being quite similar to previous efforts, the different approach here makes it a valuable contribution to the literature. I have three main areas that could be improved:

Why this model? In the Introduction, the authors make the argument that previous works were limited by using only climate model output, which could be an issue because that approach does not allow interaction between corals and climate and is constrained to the temporal and spatial resolution of the provided output. That makes sense, however it doesn't seem like the present work really improves on the previous works. The iLOVECLIM model seems quite coarse (3° ocean grids, larger than the climate model output used in at least some previous works) and the authors consider the effects of carbonate processes negligible on climate on centennial timescales. I think it is fine to present the results from this single model and explain its features and any new insights, but I am not seeing the argument that this is clearly an improvement over previous efforts in this field.

We thank the reviewer for their comments and questions.

We agree that iLOVECLIM might not be the best model to study a few future simulations, however it is a very useful model to study a large amount of future simulations, which we do here. As it is a fast model (around 700 simulated years per day) it allows us to test various sources of uncertainties and evaluate the range of possibilities.

Because it is fast and allows numerous simulations, it was also the perfect model to develop and test this first coral module embedded with in climate model. But as it is limited by the resolution and inherent simplifications of the intermediate model, we plan to also implement this module in a higher resolution GCM and evaluate the changes of carbonate production in such a model.

It would help readers if there were more explanation of the carbonate model embedded in iLOVECLIM. The results do seem highly similar to the Cornwall works, especially in the sense that the future state of reef carbonate production depends primarily on emissions scenario and heat-induced changes in coral communities and cover. The authors here note that their method of calculating carbonate production differs from the Cornwall approach of synthesizing laboratory studies. However, we never really get a clear explanation of how the carbonate module in the present study works. How was it parameterized and validated? Perhaps more emphasis could be placed on the similarity with Cornwall et al works from different approaches, but at present it is difficult to judge just how different the methods are.

As also suggested by reviewer 1, we have added more details on the coral reef module that was described and tested in Bouttes et al., 2024 so that the reader does not have to read this other paper to know how carbonate production is computed.

It should be stated that the NOAA approach to estimating bleaching is indeed an estimate and it carries substantial uncertainty. It would be difficult to do, but ideally the current error of the NOAA method for estimating bleaching could be included in uncertainty assessment run into

the future. But since the current uncertainty may not be well known, the authors should at least clearly describe this assumption.

Following the reviewer's comment, we have added some discussion in the text:

"It is thus assumed that accumulated thermal stress is the primary driver of mass bleaching events. This is of course a simplification, and the method has substantial associated uncertainty – see Klein et al. (2024) for an extensive discussion of the strengths and weaknesses of this so-called 'excess heat' threshold model (as well as those of alternatives, such as population dynamic, species distribution or ecology-evolutionary models). It should be noticed that different taxa have different responses to thermal stress and local temperature variability also plays a role (McClanahan et al., 2020). The predictive power of the method can be improved if, e.g., region-specific threshold values are adopted (DeCarlo, 2020). Here we decided to closely follow the original Coral Reef Watch methodology as it is most suitable for the level of complexity of the climatic forcings iLOVECLIM can provide. We furthermore use the original global threshold values as iCORAL does not carry any information about the reef ecosystem structure and does therefore not allow for any regional differentiation. Other stress factors besides excess heat that may lead to bleaching, such as anomalously low temperatures, anomalous nutrient concentrations, salinities etc., are already considered in the habitability criteria."

New references:

DeCarlo, T. M.: Treating coral bleaching as weather: a framework to validate and optimize prediction skill, PeerJ 8:e9449, doi: 10.7717/peerj.9449, 2020.

Klein, S. G., Roch, C. and Duarte, C. M.: Systematic review of the uncertainty of coral reef futures under climate change. Nature Communications, 15:2224, doi: 10.1038/s41467-024-46255-2, 2024

McClanahan, T. R., Darling, E. S., Maina, J. M., Muthiga, N. A., D'agata, S., Leblond, J., Arthur, R., Jupiter, S. D., Wilson, S. K., Mangubhai, S., Ussi, A. M., Guillaume, M. M. M., Humphries, A. T., Patankar, V., Shedrawi, G., Pagu, J. and Grimsditch, G.: Highly variable taxa-specific coral bleaching responses to thermal stresses, Marine Ecology Progress Series 648:135-151, doi: 10.3354/meps13402, 2020.

Along these lines, it is not clear how degree heating weeks are calculated in the present works: Is one value calculated for an entire 3x3° area? That does seem really quite coarse relative to spatial scales of marine heatwaves. At least, this should be discussed to a greater extent.

Yes, there is one DHW value computed for each grid cell. This is a limitation that is discussed in the discussion section:

"iLOVECLIM has a relatively coarse ocean resolution of 3° by 3° on the horizontal grid. While we have been able to take into account the spatial heterogeneity of the seafloor topography by adopting a subgrid parametrization, this is not possible for other variables such as temperature, as they depend on local circulation dynamics. Hence the model cannot

account for small scale features such as local temperature and $\Omega_{ar}$ changes. A consequence of this is that while large-scale changes can be evaluated with the model, local changes dependent on small-scale dynamics cannot be simulated. Such limitations could be partially addressed by the use of a higher resolution model in which the same coral module could be implemented. In addition, higher resolution models also have a higher temporal resolution, resulting in better tropical variability, which would further improve the modelled coral reef response."